# Regional anesthesia educational material utilization varies by World Bank income category: A mobile health application data study

Vanessa Moll[1,2], Edward R. Mariano[3,4], Jamie M. Kitzman[5], Vikas N. O'Reilly-Shah[6]*, Craig S. Jabaley[1]

1 Department of Anesthesiology, Emory University School of Medicine, Atlanta, Georgia, United States of America, 2 Institute of Anesthesiology, University Hospital Zurich, Zurich, Switzerland, 3 Department of Anesthesiology, Perioperative and Pain Medicine, Stanford University School of Medicine, Stanford, California, United States of America, 4 Anesthesiology and Perioperative Care Service, Veterans Affairs Palo Alto Health Care System, Palo Alto, California, United States of America, 5 Department of Pediatric Anesthesiology, Children's Healthcare of Atlanta, Emory University School of Medicine, Atlanta, Georgia, United States of America, 6 Department of Anesthesiology and Pain Medicine, University of Washington, Seattle, Washington, United States of America

* voreill@uw.edu

**Data Availability Statement:** All relevant data are within the paper and its Supporting Information files.

## Abstract

### Introduction

Regional anesthesia offers an alternative to general anesthesia and may be advantageous in low resource environments. There is a paucity of data regarding the practice of regional anesthesia in low- and middle-income countries. Using access data from a free Android app with curated regional anesthesia learning modules, we aimed to estimate global interest in regional anesthesia and potential applications to clinical practice stratified by World Bank income level.

### Methods

We retrospectively analyzed data collected from the free Android app "Anesthesiologist" from December 2015 to April 2020. The app performs basic anesthetic calculations and provides links to videos on performing 12 different nerve blocks. Users of the app were classified on the basis of whether or not they had accessed the links. Nerve blocks were also classified according to major use (surgical block, postoperative pain adjunct, rescue block).

### Results

Practitioners in low- and middle-income countries accessed the app more frequently than in high-income countries as measured by clicks. Users from low- and middle-income countries focused mainly on surgical blocks: ankle, axillary, infraclavicular, interscalene, and supraclavicular blocks. In high-income countries, more users viewed postoperative pain blocks: adductor canal, popliteal, femoral, and transverse abdominis plane blocks. Utilization of the app was constant over time with a general decline with the start of the COVID-19 pandemic.

**Funding:** This research received no specific grant from any funding agency in the public, commercial, or not-for-profit sectors. The Emory University Department of Anesthesiology supported this work with non-clinical time for the authors. The funders had no role in study design, data collection and analysis, decision to publish, or preparation of the manuscript.

**Competing interests:** All authors declare no support from any organization for the submitted work; no financial relationships with any organizations that might have an interest in the submitted work in the previous three years; no other relationships or activities that could appear to have influenced the submitted work. The app was initially released in 2011 by Vikas O'Reilly-Shah with advertising in the free version and a paid companion app to remove the ads. The app intellectual property was transferred to Emory University in 2015 and advertisements were subsequently removed. The companion app to remove ads made freely is available for legacy users not updating to the ad-free version. Following review by the Emory University Research Conflict of Interest Committee, Vikas O'Reilly-Shah has been released from any conflict of interest management plan or oversight. This does not alter our adherence to PLOS ONE policies on sharing data and materials.

## Conclusion

The use of an in app survey and analytics can help identify gaps and opportunities for regional anesthesia techniques and practices. This is especially impactful in limited-resource areas, such as lower-income environments and can lead to targeted educational initiatives.

## Introduction

According to estimates from the Lancet Commission on Global Surgery, 5 billion people lack access to safe and affordable surgery and anesthesia, accounting for 28% of the global burden of disease [1]. Regional anesthesia can both reduce perioperative morbidity and improve the availability of safe anesthesia in austere settings [2]. Advantages include excellent operating conditions, profound analgesia without sedation, stable hemodynamics, reduced need for other anesthetics and oxygen, and rapid recovery [3–5]. Forgoing general anesthesia confers indirect benefits, such as avoidance of airway manipulation and the side effects of anesthetic drugs, which may favorably impact the allocation of resources during postanesthesia recovery.

Despite these benefits, the adoption of these regional techniques is limited, even in high-income countries [6]. Nearly half of the Disease Control Priorities-3 core procedures can be performed under neuraxial or regional anesthesia [7]. However, a lack of physician instructors, essential drugs, and required equipment have limited the teaching and implementation of regional anesthesia techniques in low and middle-income countries (LMIC) [8]. Limited data are available to gauge the interest in, or adoption of, peripheral nerve block (PNB) techniques in these environments.

Worldwide adoption of mobile health applications (apps) via smartphone technology offers an opportunity to assess global interest in regional anesthetic techniques. A widely adopted anesthesiology clinical decision support app called "Anesthesiologist" has previously demonstrated disproportionate utilization in LMICs [9]. The app has been used to crowdsource information about factors influencing choice of neuromuscular blockade reversal [10], adverse drug events [11], and the use of a critical events checklist [12]. The use of the app has served as a proxy for global surgical case volumes, and utilization metadata have been used to track the impact of COVID-19 on these volumes [13]. This app contains links to online educational resources related to the performance of a variety of ultrasound-guided peripheral nerve blocks.

There is a paucity of data on the use of regional anesthesia techniques in LMICs given well-recognized barriers to relevant data gathering, reporting, and scholarly dissemination. Estimating the level of interest in regional anesthesia techniques and the frequency of their clinical application may help local and international organizations formulate a basic needs assessment, propose targeted educational interventions, and potentially expand regional anesthesia availability and utilization.

By examining user utilization of nerve block educational resources within the "Anesthesiologist" app, we aimed to characterize global interest in regional anesthesia stratified by World Bank country income level.

## Methods

This project was approved by the Emory University Institutional Review Board (Atlanta, GA, USA; IRB# 00082571), and participants provided written informed consent electronically.

## Outcomes

Our primary outcome was the frequency of accessing nerve block videos in relation to the World Bank country income levels. Educational videos were classified into 3 groups: peripheral surgical nerve blocks (ankle block, axillary, infraclavicular, interscalene, supraclavicular blocks), postoperative pain nerve blocks (adductor canal, femoral, popliteal, transverse abdominis plane blocks) and rescue nerve blocks (median, radial, ulnar nerve blocks).

## Data collection

The Android platform app Anesthesiologist has been freely available from the Google Play store since 2011. It is a basic single screen age- and weight-based calculator that presents airway equipment information, normal ranges for physiological parameters, and drug dose calculations (details of the app and a web link can be found in S1 File). Links to publicly available educational resources are also provided, including externally sited educational videos related to the performance of twelve different peripheral nerve blocks (S1 Fig). The app was only available for Android devices. Android's mobile operating system market share in August 2020 was 74.25% globally with higher penetration rates in LMICs, such as those on the African continent or in and South America [14].

Via the open-source Survalytics module integrated into the app, we collected basic user demographic information via a survey instrument and analytics about in-app behavior as has been previously described [9, 15] and in the Supplement (S1 Table, S2 and S3 Files). These data were stored in a cloud database provided by Amazon Web Services (Amazon Web Services, Inc., Seattle, WA). Survey data include basic clinically relevant demographics: professional role, length of time in practice, practice size, practice model, and practice environment. Highly detailed and discretized anonymous app utilization data are collected, including timestamps, location, and in-app activities. For the present study, we analyzed data from December 15, 2015 through April 18, 2020. Recorded app utilization data included navigation to any external site linked within the app.

## Data processing and descriptive analysis

Data from the app was processed using RStudio v3.6.3 (R Core Team, Vienna, Austria) [16–19]. World Bank classification of country income level was made based on the publicly available classification as of May 2020 [20]. According to the World Bank Atlas methodology, economies are classified according to gross national income (GNI) per capita [21]. Each unique access to a nerve block link contained country of origin metadata, which was categorized according to World Bank criteria. Additionally, each unique user's primary country was characterized according to the country from which they accessed the app most frequently; this was also categorized according to World Bank criteria. User demographic data and nerve block link utilization were tabulated and summarized. For these tables, nerve block link utilization was based on raw counts of link activation within the app (users may have accessed nerve block links more than once). A choropleth was generated to depict the count of unique users accessing nerve block information, relative to the total number of unique users per country, using the tmap package for R [22].

## Results

Data were collected from 139,619 consenting users during the study period. Self-reported demographics obtained via survey instrument are reported in Table 1. The provider level (physician, trainee, anesthesia assistant etc.) is described in Table 1 with 47.3% being physicians

**Table 1. Demographics of app users globally.**

|  | N (%) |
|---|---|
| **App Users** | 139 619* |
| **Professional Role** |  |
| Physician Attending/Consultant | 23 033 (26.3) |
| Physician Resident/Fellow/Registrar | 18 360 (21.0) |
| CRNA or AA | 21 712 (24.8) |
| CRNA or AA Trainee | 4 060 (4.6) |
| Technically Trained in Anesthesia | 2 334 (2.7) |
| Anesthesia Technician | 5 392 (6.2) |
| Medical Student | 4 047 (4.6) |
| Nurse | 2 971 (3.4) |
| Paramedic/EMT | 2 053 (2.3) |
| Respiratory Therapist | 664 (0.8) |
| Pharmacist | 782 (0.9) |
| Other | 1 356 (1.5) |
| Not Medical Provider | 852 (1.0) |
| **Length of Practice (mean, in years)** | **13.07 (13.47)** |
| **Practice Model** |  |
| Physician only | 11 169 (30.9) |
| Physician supervised, anesthesiologist on site | 15 076 (41.7) |
| Physician supervised, non-anesthesiologist on site | 3 246 (9.0) |
| Physician supervised, no physician on site | 2 057 (5.7) |
| No physician supervision | 2 521 (7.0) |
| Not an anesthesia provider | 2 063 (5.7) |
| **Practice Type** |  |
| Private clinic or office | 7 689 (20.1) |
| Local health clinic | 3 525 (9.2) |
| Ambulatory surgery center | 2 437 (6.4) |
| Small community hospital | 4 633 (12.1) |
| Large community hospital | 10 393 (27.2) |
| Academic department/University hospital | 9 575 (25.0) |
| **Practice size** |  |
| I am the only practitioner for large area | 12 749 (28.0) |
| One of several practitioners in the area | 8 790 (19.3) |
| Group practice 1–5 members | 6 068 (13.3) |
| Group practice 5–10 members | 4 823 (10.6) |
| Group practice 10–25 members | 4 468 (9.8) |
| Group practice 25–50 members | 3 539 (7.8) |
| Group practice >50 members | 5 048 (11.1) |

Abbreviations: CRNA, certified registered nurse anesthetist, AA, anesthesiologist assistant, EMT, emergency medical technician.

*Discrepancy between total N app users and demographic information due to app users not completing the demographic questionnaire.

(attendings and trainees combined) and 29.4% certified registered nurse anesthetists (CRNA) or anesthesiologist assistants (AA) or respective trainees. The peripheral nerve block tutorials were accessed 30,471 times from 6 continents (see Figs 1 and 2).

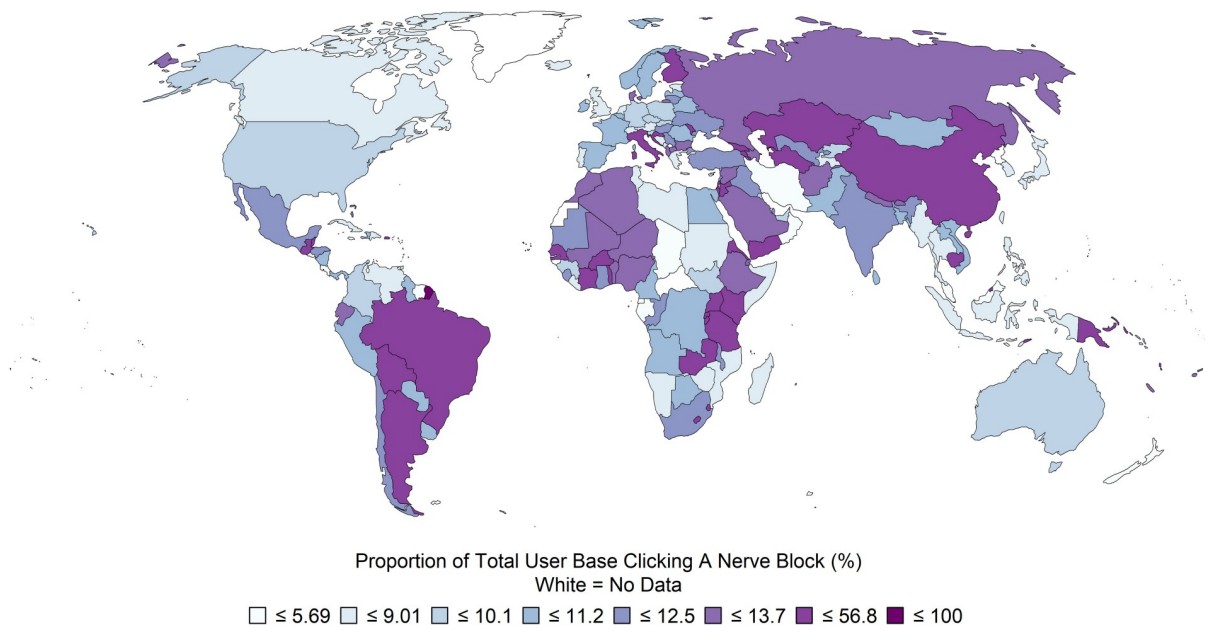

Proportion of Total User Base Clicking A Nerve Block (%)
White = No Data

□ ≤ 5.69 □ ≤ 9.01 ☐ ≤ 10.1 ☐ ≤ 11.2 ▨ ≤ 12.5 ▨ ≤ 13.7 ▨ ≤ 56.8 ■ ≤ 100

**Fig 1. Proportion of total user base clicking on nerve block links.**

## Primary outcome

Practitioners in LMIC accessed the linked nerve block videos more frequently (22,062 clicks) than in HIC (8,409 clicks).

Differences were found in nerve block categories per World Bank country income level with surgical blocks receiving more clicks in low (1484/61.5%), in lower middle (6457/66.1%), and in upper middle-income countries (6592/66.8%) than in HIC (4822/57.3%). Postoperative pain blocks were clicked at a higher percentage (2715/32.3%) in HIC compared to low (521/21.6%), lower middle (2090/21.4%), and upper middle-income countries (2112/21.4%). Rescue blocks received the least clicks across income levels, but the level of interest was highest in LIC with 409 (16.9%) clicks (Table 2).

The interest in various types of nerve blocks broken down by country income level is shown in Table 3. The interscalene block received the most clicks (8021) with 6.5%, 31%,

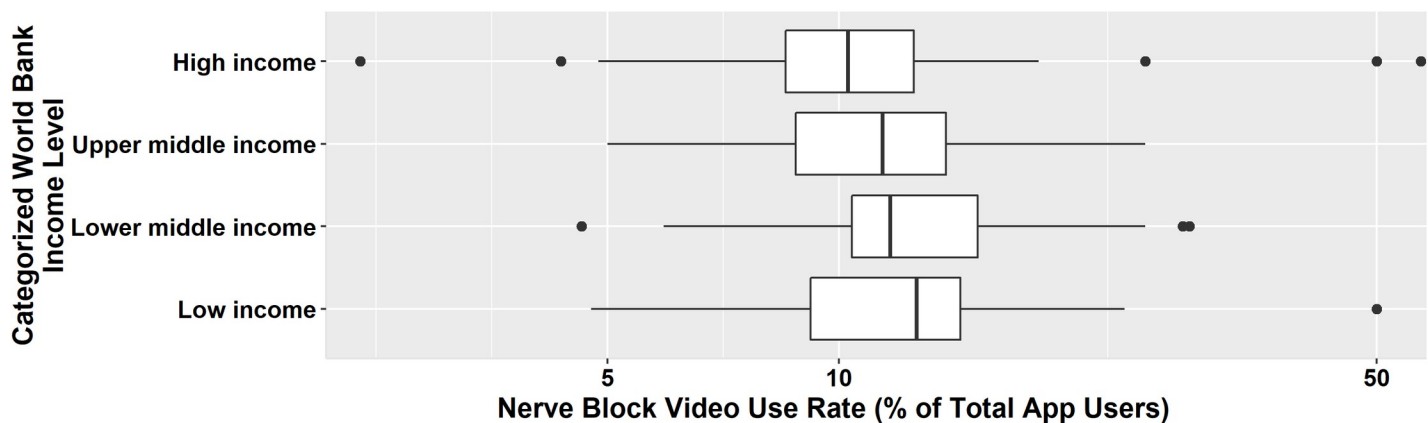

**Fig 2. Proportion of app user base looking at nerve block videos categorized by World Bank income level.**

**Table 2. Interest (as clicks) in different nerve blocks categorized as postoperative pain, rescue and surgical nerve blocks by income level.** **A:** Percentages describe nerve block categories (row percent). **B:** Percentages describe income levels (column percent).

| Nerve Block Groups | N | Low Income | Lower Middle Income | Upper Middle Income | High Income |
|---|---|---|---|---|---|
| A | | | | | |
| Postoperative Pain | 7438 | 521 (7%) | 2090 (28.1%) | 2112 (28.4%) | 2715 (36.5%) |
| Rescue | 3678 | 409 (11.1%) | 1228 (33.4%) | 1169 (31.8%) | 872 (23.7%) |
| Surgical | 19355 | 1484 (7.7%) | 6457 (33.4%) | 6592 (34.1%) | 4822 (24.9%) |
| B | | | | | |
| Postoperative Pain | 7438 | 521 (21.6%) | 2090 (21.4%) | 2112 (21.4%) | 2715 (32.3%) |
| Rescue | 3678 | 409 (16.9%) | 1228 (12.6%) | 1169 (11.8%) | 872 (10.4%) |
| Surgical | 19355 | 1484 (61.5%) | 6457 (66.1%) | 6592 (66.8%) | 4822 (57.3%) |

36.7% and 25.8% of total clicks for nerve blocks coming from low, lower middle, upper middle, and HIC, respectively. Generally, rescue blocks were the least accessed with 3768 clicks; of these, the ulnar block received the least interest with 1007 total clicks divided into 3.9% of total clicks from low, 3.2% from lower middle, 3.6% from upper middle and 3% from HIC.

**Table 3. Interest (as clicks) in different nerve blocks by income level.** Percentages are described by nerve blocks. **A:** Percentages describe nerve block categories (row percent). **B:** Percentages describe income levels (column percent).

| Type of Nerve Block | N | Low Income | Lower Middle Income | Upper Middle Income | High Income |
|---|---|---|---|---|---|
| A | | | | | |
| Adductor canal | 1272 | 75 (5.9%) | 323 (25.4%) | 312 (24.5%) | 562 (44.2%) |
| Ankle block | 2325 | 180 (7.7%) | 997 (42.9%) | 613 (26.4%) | 535 (23%) |
| Axillary | 2982 | 273 (9.2%) | 866 (29%) | 1004 (33.7%) | 839 (28.1%) |
| Femoral | 2188 | 145 (6.6%) | 660 (30.2%) | 615 (28.1%) | 768 (35.1%) |
| Infraclavicular | 1867 | 200 (10.7%) | 595 (31.9%) | 637 (34.1%) | 435 (23.3%) |
| Interscalene | 8021 | 525 (6.5%) | 2484 (31%) | 2940 (36.7%) | 2072 (25.8%) |
| Median | 1041 | 161 (15.5%) | 368 (35.4%) | 286 (27.5%) | 226 (21.7%) |
| Popliteal | 1497 | 123 (8.2%) | 406 (27.1%) | 451 (30.1%) | 517 (34.5%) |
| Radial | 1630 | 155 (9.5%) | 550 (33.7%) | 532 (32.6%) | 393 (24.1%) |
| Supraclavicular | 4160 | 306 (7.4%) | 1515 (36.4%) | 1398 (33.6%) | 941 (22.6%) |
| TAP | 2481 | 178 (7.2%) | 701 (28.3%) | 734 (29.6%) | 868 (35%) |
| Ulnar | 1007 | 93 (9.2%) | 310 (30.8%) | 351 (34.9%) | 253 (25.1%) |
| B | | | | | |
| Adductor canal | 1272 | 75 (3.1%) | 323 (3.3%) | 312 (3.2%) | 562 (6.7%) |
| Ankle block | 2325 | 180 (7.5%) | 997 (10.2%) | 613 (6.2%) | 535 (6.4%) |
| Axillary | 2982 | 273 (11.3%) | 866 (8.9%) | 1004 (10.2%) | 839 (10%) |
| Femoral | 2188 | 145 (6%) | 660 (6.8%) | 615 (6.2%) | 768 (9.1%) |
| Infraclavicular | 1867 | 200 (8.3%) | 595 (6.1%) | 637 (6.5%) | 435 (5.2%) |
| Interscalene | 8021 | 525 (21.7%) | 2484 (25.4%) | 2940 (29.8%) | 2072 (24.6%) |
| Median | 1041 | 161 (6.7%) | 368 (3.8%) | 286 (2.9%) | 226 (2.7%) |
| Popliteal | 1497 | 123 (5.1%) | 406 (4.2%) | 451 (4.6%) | 517 (6.1%) |
| Radial | 1630 | 155 (6.4%) | 550 (5.6%) | 532 (5.4%) | 393 (4.7%) |
| Supraclavicular | 4160 | 306 (12.7%) | 1515 (15.5%) | 1398 (14.2%) | 941 (11.2%) |
| TAP | 2481 | 178 (7.4%) | 701 (7.2%) | 734 (7.4%) | 868 (10.3%) |
| Ulnar | 1007 | 93 (3.9%) | 310 (3.2%) | 351 (3.6%) | 253 (3%) |

Abbreviations: TAP, transverse abdominis block

## Discussion

In this investigation examining anesthesiology clinical reference app utilization data from a worldwide user base, different levels of interest for nerve blocks according to World Bank income were found. With in-app clicks as the proxy measure, LMIC users demonstrated greater interest in surgical peripheral nerve blocks (ankle block, axillary, infraclavicular, interscalene, supraclavicular blocks) as compared to users from HIC.

Users from HIC showed most interest in the blocks mainly utilized to treat postoperative pain (adductor canal, femoral, popliteal, transverse abdominis plane blocks). Users from LMIC more frequently accessed regional anesthetic reference materials than did those from HIC. The COVID-19 pandemic overlapped with our app study period and was associated with substantial app usage reductions. In a separate study, app data provided a proxy for surgical case volumes, and could therefore be used as a real-time monitor of the impact of COVID-19 on surgical capacity [13]. In the literature, the largest studies describing anesthesia practice including regional techniques in LMIC are a large cohort study describing Médecins Sans Frontières activity during a 6-year period and a register study analyzing French anesthetic activity in a deployed military setting [2, 5]. However, both these studies describe transplanted (practitioners from mostly HIC working in low-resourced settings) and not local practice patterns.

A shortage of resources such as monitors, oxygen or personnel emphasize the importance of the availability of regional anesthesia techniques in austere settings [5, 23]. This is reflected in the first World Federation of Societies of Anesthesiologists sponsored East African regional anesthesia and acute pain medicine fellowship. The results of this study may aid in promoting and tailoring regional anesthesia techniques to actual needs in LMIC. We found a higher interest in surgical peripheral nerve blocks in LMIC, a majority of these being upper extremity nerve blocks. The lack of interest in lower extremity blocks might be of practical nature. There is a relatively high capacity for spinal anesthesia in LMIC [24] and hence this could be the preferred method of regional anesthesia for below the umbilicus surgery [6]. Our category postoperative pain blocks included lower extremity and TAP blocks, which cover areas potentially amenable to be covered by spinal anesthesia. Although speculative, the increased affordability of ultrasound devices combined with a growing interest in regional anesthesia techniques may lead to an increase in PNB utilization at the expense of neuraxial anesthesia in the future [25].

Besides the lack of equipment and disposables, adequate training poses a major driving factor in the application of regional anesthesia in LMIC [8, 26, 27]. This dataset outlining high yield peripheral nerve blocks might inform the development of a standardized regional anesthesia curriculum. Recently, competence in performing a small number of nerve blocks covering the majority of surgical procedures was promoted to allow greater access for patients to regional anesthesia [28]. The proposed blocks were anatomically divided in upper limb (infraclavicular and interscalene nerve blocks), lower limb (femoral, popliteal and adductor canal nerve blocks) and trunk blocks (erector spinae and rectus sheath blocks). The ability to perform high yield nerve blocks in limited-resource settings might allow more patients to undergo safe anesthesia and surgery. Data from this study can aid in informing the development of future regional anesthesia curriculum standards in LMICs.

### The use of social media in regional anesthesia education

This study is utilizing crowd-sourced educational videos describing different ultrasound guided nerve blocks. Particularly in recent years, the use of social media and the advent of free online medical education (FOAM) has grown substantially in medicine. Specifically, in regional anesthesia, the benefits and challenges of FOAM are described in detail [29, 30]. Although these crowd-sourced resources need to be met with a healthy amount of skepticism,

the sharing of videos and images especially in ultrasound guided procedures has led to improved education in regional anesthesia [29]. In anesthesiology, the use of FOAM and digital platforms has allowed for greater accessibility, portability and flexibility in medical education [31]. A recent review even argued that the "future of medical education is social (media)" [32]. Crowd-sourced educational materials are not a substitute for textbooks and clinical training but are becoming a bigger part of the education for both novice and experienced regional anesthesiologists with benefits including global interaction and knowledge translation within the specialty. With the growing FOAM movement, there is now an unprecedented ability to share information beyond traditional venues (e.g., conferences or publications) which might have an impact especially in low-resourced countries [33–36].

### Strength and limitations

Our methodology offers one potential means by which to assess global interest in regional anesthesia techniques. Mobile technology has been widely adopted in LMIC [37], and examining the utilization of mobile health information offers previously unassessed insights into clinical practice. One strength of using an app-based platform to explore related questions is the ability to develop and deploy survey instruments for any number of topics while also collecting app utilization data. In overstretched and under-resourced settings, such as LMIC, the collection of healthcare-related data with mobile technology presents both opportunities and challenges [38, 39]. Gathering data about health or healthcare information-seeking via mobile health apps is feasible, timely, and cost-effective to deploy at scale [40]. Mobile networks are advantageous in areas with limited landline infrastructure. For instance, in 2015 there were 5.5 billion mobile phone subscriptions in developing countries, representing nearly 92 subscriptions per 100 inhabitants [37]. Mobile applications therefore offer an opportunity to better understand clinical practice, information-seeking behavior, and other behaviors relevant to health services research within LMIC.

Our application is exclusive to the Android platform, which likely serves to introduce selection bias. However, Android is estimated to have greater global penetration compared to other mobile operating systems, which may serve to capture both more users and a greater proportion of users outside of high-income countries [14]. We examined user behavior of only one mobile health application, and our findings are therefore specific to this particular subset of the greater healthcare workforce. As such, the generalizability of our findings beyond this subset is limited. Among users of the application, differences in participant income, consistency or quality of internet access, parallel utilization of offline reference material, local practice patterns, or other unaccounted-for variables are all important factors that may serve to introduce confounding. Within the context of these important limitations, our findings offer novel insight into clinical practice patterns that might not otherwise be possible on a global scale via alternative methodologies.

Regarding app utilization itself, clicks to nerve block videos demonstrated interest in a certain block, but completion of the video or performance of the actual nerve block could not be ascertained. Additionally, some of the videos demonstrate ultrasound-guided techniques, technology that may not be available to some practitioners. Finally, our classification in surgical, postoperative pain, and rescue blocks leaves room for discussion. Certainly, femoral or popliteal blocks can—either alone or in combination—be utilized as a sole anesthetic in localized surgeries. However, in the authors' experience, these blocks are most commonly used as postoperative analgesic therapy.

### Conclusions

In conclusion, there may be a difference in the interest level of different nerve blocks according to a country's income level. Surgical peripheral nerve blocks received the most clicks in LMIC,

and the highest interest in rescue blocks was found in low-income countries. A higher level of interest in upper extremity blocks was seen in LMIC. This study is a first step in elucidating levels of interest in peripheral nerve blocks globally. Data can be used to define standards and basic blocks that can be performed safely in low resource settings. Next steps could include separating the countries regionally, as there would likely be differences in metropolitan areas compared to rural areas due to the availability of resources. This data could then be used to design features within the app to bridge knowledge deficits where no other resources are available. Future studies should elucidate if features in mobile technology can provide meaningful educational resources to promote safe regional anesthesia practices.

## Supporting information

**S1 Fig. Screenshot of the app.**
(PDF)

**S1 File. Description of the app, including links to the app.**
(PDF)

**S2 File. Survalytics detailed description.**
(PDF)

**S3 File. Mobile healthcare app study database schema.**
(PDF)

**S1 Table. Survey for collection of basic demographic information.**
(PDF)

**S2 Table. Raw data used to generate choropleth.**
(PDF)

## Acknowledgments

We would like to thank Huseyin Hamid Tunceroglu for his efforts working on this project during early stages of development as a pediatric anesthesiology fellow. We would like to thank Emory University Department of Anesthesiology, the University of Washington Department of Anesthesiology & Pain Medicine and the Institute of Anesthesiology at the University Hospital Zurich, Switzerland for their generous support of the authors' time to perform this work.

## Author Contributions

**Conceptualization:** Vikas N. O'Reilly-Shah.

**Data curation:** Vikas N. O'Reilly-Shah.

**Formal analysis:** Vikas N. O'Reilly-Shah.

**Methodology:** Vikas N. O'Reilly-Shah.

**Project administration:** Vikas N. O'Reilly-Shah.

**Validation:** Vikas N. O'Reilly-Shah.

**Writing – original draft:** Vanessa Moll.

**Writing – review & editing:** Vanessa Moll, Edward R. Mariano, Jamie M. Kitzman, Vikas N. O'Reilly-Shah, Craig S. Jabaley.

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
