## [Decision Letter · Decision Letter 0]

10 Sep 2020

PONE-D-20-25898

Evaluating worldwide interest in regional anesthesia educational resources: an observational study of mobile health application data

PLOS ONE

Dear Dr. Moll,

Thank you for submitting your manuscript to PLOS ONE. After careful consideration, we feel that it has merit but does not fully meet PLOS ONE’s publication criteria as it currently stands. Therefore, we invite you to submit a revised version of the manuscript that addresses the points raised during the review process.

This study describes the pattern of smart phone application use by healthcare providers in low to middle income countries.  The authors studied the type and frequency of regional anesthesia blocks accessed in an application called “Anesthesiologist”.  It is a well conducted survey and has considerable information.  But, the clinical application of the data and related quality improvement interventions are unclear.  The reviewers have provided important input and I ask the authors to address each question/comment.

I am providing editorial comments about limitations of this study that should be addressed.  It was not clear what specific problems the investigators are studying, who this is important to and what gap in knowledge this study fills.  These points are essential building blocks that should be clearly articulated in the Introduction of every scientific paper so the readers can understand why the study was done.  The contextual framing for this study is therefore underdeveloped.  Readers need this information to  understand how this study will advance their knowledge and improve medical care.  Can the authors please revise the paper to make these points clear. 

There are a number of biases in this study that have not been fully explored.  I share the critique brought up by the reviewers regarding the use of Android as the sole source of the study data.  It is also possible that providers used other Android applications.  The authors examined a study sample and not a census.  All samples need to be well defined; i.e. how the sample relates the whole population.  The authors would have to cite how many healthcare providers from LMIC use Android compared to other types of smart phones or even those with no access to this technology.  How do these populations compare to each other?   Do they have the same resources and approach to treatment?  It is unlikely this data is available so the authors should acknowledge this as an important limitation to their study. 

It is problematic the authors are studying a section of the healthcare population that they are unable to define in comparison to the whole.  Other important sources of bias the authors have not discussed thoroughly include a lack of information on how many healthcare providers with potential access did not use this resource.  Once again, I anticipate this information may be almost impossible to access but the issue deserves more attention.  The authors are obliged to explain that the information in this study cannot be generalized and limits the clinical utility of the data. 

Can the authors provide some type of link to the “Anesthesiologist”?  The authors should include a discussion about the “evidence/validity” for crowd sourced educational materials.  Is this application the Wikipedia of anesthesia practice?  Is there any peer review of the materials? 

The discussion contains a considerable amount of speculation and theorizing based on little data.  I ask the authors to revise this section to explore the strengths and weaknesses of their data. What do their findings mean?   It is not helpful to speculate that certain blocks may be used because of more traffic accidents when there is no data in this or other studies to support this statement.  But the authors can present this as a question that needs further evaluation in another study.

We look forward to receiving your revised manuscript.

Kind regards,

Mercedes Susan Mandell, MD PhD

Academic Editor

PLOS ONE

Journal Requirements:

2. We note that Figure 1 in your submission contain map images which may be copyrighted. All PLOS content is published under the Creative Commons Attribution License (CC BY 4.0), which means that the manuscript, images, and Supporting Information files will be freely available online, and any third party is permitted to access, download, copy, distribute, and use these materials in any way, even commercially, with proper attribution. For these reasons, we cannot publish previously copyrighted maps or satellite images created using proprietary data, such as Google software (Google Maps, Street View, and Earth). For more information, see our copyright guidelines: http://journals.plos.org/plosone/s/licenses-and-copyright.

2.1.    You may seek permission from the original copyright holder of Figure 1 to publish the content specifically under the CC BY 4.0 license. 

2.2.    If you are unable to obtain permission from the original copyright holder to publish these figures under the CC BY 4.0 license or if the copyright holder’s requirements are incompatible with the CC BY 4.0 license, please either i) remove the figure or ii) supply a replacement figure that complies with the CC BY 4.0 license. Please check copyright information on all replacement figures and update the figure caption with source information. If applicable, please specify in the figure caption text when a figure is similar but not identical to the original image and is therefore for illustrative purposes only.

Reviewers' comments:

Reviewer's Responses to Questions

**Comments to the Author**

1. Is the manuscript technically sound, and do the data support the conclusions?

Reviewer #1: Partly

Reviewer #2: Partly

Reviewer #3: Yes

2. Has the statistical analysis been performed appropriately and rigorously? 

Reviewer #1: I Don't Know

Reviewer #2: Yes

Reviewer #3: Yes

3. Have the authors made all data underlying the findings in their manuscript fully available?

Reviewer #1: Yes

Reviewer #2: Yes

Reviewer #3: Yes

4. Is the manuscript presented in an intelligible fashion and written in standard English?

Reviewer #1: Yes

Reviewer #2: Yes

Reviewer #3: Yes

5. Review Comments to the Author

Reviewer #1: A timely and well thought out study based on digital technology usage patterns. The following points however are worth considering.

Key message:

Title:

A distribution of the countries/ regions from which responses were elicited would have indicated that this was truly a study which reflected worldwide interest. Since an income based analysis has been carried out the title does not adequately reflect that this was primarily an analysis based on income levels and app utilization.

Evidence & examples:

Primary outcome:

Practitioners in LMIC accessed the app more frequently (22,062 clicks) than in HIC (8,409 140 clicks). – It appears that this refers to the usage of the App. Since the App provides other information eg. Drug dose calculation, it may be pertinent to compare access of Nerve block related material eg. tutorials specifically.

Discussion:

“LMIC users had the highest in-app clicks for surgical peripheral nerve blocks”

Since the results maybe used for designing educational tools based on the unique requirements of each group, it might be useful to consider size of practitioner population in each of these categories (LMIC, HIC) and give appropriate weightage when considering the utilization of the App and its links as well as in arriving at conclusions.

“Postoperative pain blocks were clicked at a higher percentage (2715/32.3%) in HIC compared to low (521/21.6%), lower middle (2090/21.4%), and 145 upper middle-income countries (2112/21.4%)”.

Maybe pertinent to consider the effect of confounding factors such as local policy relating to post operative pain relief that may have produced the above result.

The level of significance of the difference in App utilization between different income regions is not clearly explained.

Additional comments:

Reviewer #2: the data collection done can be interpreted as being one sided. The authors discuss how their collection of data may have answered their question, but they do fail to report on issues that may make the data collected less significant. Although they discuss the use of an app (that apparently just has links to youtube videos) they fail to simply comment and discuss the fact that it is an android app. One can assume that there is going to be a variance in the HIC to the LIC in terms of the use of android devices to non android devices. Does the app only work on android devices or can it be used on non-android devices as well? If that is the case, the introduction can lead to a little confusion. If it is that the app cannot, then the data might be slightly different if that unmentioned data is included. Meaning, those without androids how to the learn and what are they searching of if looking up these videos on youtube directly? However that is not discussed and it is not entirely clear, in my eyes. In many HIC, like the U.S. and Eastern Europe, non-android devices may be utilized more frequently. Also, five years of data collection with todays technology may have data that may favor prior years to later years and vice-versa. It is nice that the effect of COVID-19 is discussed, but is the data discovered simply because case loads were down or because people stopped looking or both or other? Would be interesting to comment about in the discussion. I think they do answer their question of whether such an app can provide useful data and potentially be utilized for survey collection, however.

Reviewer #3: well written ,insightful , the material is very relevant to current standards of practice in both academic and private group setting. I am recommending this material for publishing with no reservations

6. PLOS authors have the option to publish the peer review history of their article (what does this mean?). If published, this will include your full peer review and any attached files.

Reviewer #1: **Yes: **MAHIMAN BHAGYA GUNETILLEKE

Reviewer #2: No

Reviewer #3: **Yes: **Andrei V Kopylov, MD

---

## [Author Response · Author response to Decision Letter 0]

28 Oct 2020

Revision Notes:

Editors comment 1: I am providing editorial comments about limitations of this study that should be addressed. It was not clear what specific problems the investigators are studying, who this is important to and what gap in knowledge this study fills. These points are essential building blocks that should be clearly articulated in the Introduction of every scientific paper so the readers can understand why the study was done. The contextual framing for this study is therefore underdeveloped. Readers need this information to understand how this study will advance their knowledge and improve medical care. Can the authors please revise the paper to make these points clear. 

Thank you for the opportunity to clarify. We rewrote the Introduction (including abstract introduction) to reflect these suggestions.

Editors comment 2: There are a number of biases in this study that have not been fully explored. I share the critique brought up by the reviewers regarding the use of Android as the sole source of the study data. It is also possible that providers used other Android applications. The authors examined a study sample and not a census. All samples need to be well defined; i.e. how the sample relates the whole population. The authors would have to cite how many healthcare providers from LMIC use Android compared to other types of smart phones or even those with no access to this technology. How do these populations compare to each other? Do they have the same resources and approach to treatment? It is unlikely this data is available so the authors should acknowledge this as an important limitation to their study. It is problematic the authors are studying a section of the healthcare population that they are unable to define in comparison to the whole. Other important sources of bias the authors have not discussed thoroughly include a lack of information on how many healthcare providers with potential access did not use this resource. Once again, I anticipate this information may be almost impossible to access but the issue deserves more attention. The authors are obliged to explain that the information in this study cannot be generalized and limits the clinical utility of the data. 

Thank you for these suggestions, and we agree that these are critical considerations that warrant clarification in the manuscript to better frame our work. We incorporated the following into the manuscript: 

 Gathering data about health or healthcare information-seeking via mobile health apps is feasible, timely, and cost-effective to deploy at scale [40]. Mobile networks are advantageous in areas with limited landline infrastructure. For instance, in 2015 there were 5.5 billion mobile phone subscriptions in developing countries, representing nearly 92 subscriptions per 100 inhabitants [37]. Mobile applications therefore offer an opportunity to better understand clinical practice, information-seeking behavior, and other behaviors relevant to health services research within LMIC.

Our application is exclusive to the Android platform, which likely serves to introduce selection bias. However, Android is estimated to have greater global penetration compared to other mobile operating systems, which may serve to capture both more users and a greater proportion of users outside of high-income countries [14]. We examined user behavior of only one mobile health application, and our findings are therefore specific to this particular subset of the greater healthcare workforce. As such, the generalizability of our findings beyond this subset is limited. Among users of the application, differences in participant income, consistency or quality of internet access, parallel utilization of offline reference material, local practice patterns, or other unaccounted-for variables are all important factors that may serve to introduce confounding. Within the context of these important limitations, our findings offer novel insight into clinical practice patterns that might not otherwise be possible on a global scale via alternative methodologies.

Editors comment 3: Can the authors provide some type of link to the “Anesthesiologist”? The authors should include a discussion about the “evidence/validity” for crowd sourced educational materials. Is this application the Wikipedia of anesthesia practice? Is there any peer review of the materials? 

We added the following to the supplement:

Description of the app Anesthesiologist

The app “Anesthesiologist” was designed to help quickly calculate adult and pediatric anesthesia related information like common drug dosing or airway related information such as endotracheal tube size. The app was written solely to be used as a helpful adjunct for professionally trained physicians and practitioners otherwise experienced in airway management and drug administration and dosing. There is no peer review of the app in the classical sense; the app functions much as a calculator and not as guidelines. The anesthesiologist app does provide crowd sourced educational material in the form of links to nerve blocks available publicly on YouTube (videos peer-reviewed by the app’s author) and it crowdsources information by asking professionals to voluntarily fill out surveys. 

The app can be found here: https://play.google.com/store/apps/details?id=com.shahlab.anesthesiologist&hl=en_US

We also added a section discussing crowd-sourced education or free online medical educational resources in the discussion:

The use of social media in regional anesthesia education

This study is utilizing crowd-sourced educational videos describing different ultrasound guided nerve blocks. Especially, in recent years the use of social media and the advent of free online medical education (FOAM) has grown substantially in medicine and specifically in regional anesthesia with its benefits and challenges described in detail elsewhere [28,29]. Although these crowd-sourced resources need to be met with a healthy amount of scepticism, the sharing of videos and images especially in ultrasound guided procedures has led to improved education in regional anesthesia [28]. In anesthesiology, the use of FOAM and digital platforms has allowed for greater accessibility , portability and flexibility in medical education [30]. A recent review even argued that the “future of medical education is social (media)” [31]Crowd-sourced educational materials are not a substitute for textbooks and clinical training, but are becoming a bigger part of the education for both novice and experienced regional anesthesiologists with benefits such as global interaction and knowledge translation within the specialty. With the growing FOAM movement there is now an unprecedented ability to share information beyond traditional venues (e.g. conferences or publications) which might have an impact especially in low-resourced countries [32–35].

Editors comment 4: The discussion contains a considerable amount of speculation and theorizing based on little data. I ask the authors to revise this section to explore the strengths and weaknesses of their data. What do their findings mean? It is not helpful to speculate that certain blocks may be used because of more traffic accidents when there is no data in this or other studies to support this statement. But the authors can present this as a question that needs further evaluation in another study.

Thank you for these suggestions. We eliminated our reference to road traffic accidents and realigned the discussion.

Reviewer 1, comment 1: A distribution of the countries/ regions from which responses were elicited would have indicated that this was truly a study which reflected worldwide interest. Since an income based analysis has been carried out the title does not adequately reflect that this was primarily an analysis based on income levels and app utilization.

We changed the title to the following: Evaluating global interest in regional anesthesia educational resources based on income levels: an observational study of mobile health application data

Reviewer 1, comment 2: Practitioners in LMIC accessed the app more frequently (22,062 clicks) than in HIC (8,409 140 clicks). – It appears that this refers to the usage of the App. Since the App provides other information eg. Drug dose calculation, it may be pertinent to compare access of Nerve block related material eg. tutorials specifically.

Thank you for this comment we clarified in the Results: Practitioners in LMIC accessed the linked nerve block videos more frequently (22,062 clicks) than in HIC (8,409 clicks). 

Reviewer 1, comment 3: “LMIC users had the highest in-app clicks for surgical peripheral nerve blocks” Since the results may be used for designing educational tools based on the unique requirements of each group, it might be useful to consider size of practitioner population in each of these categories (LMIC, HIC) and give appropriate weightage when considering the utilization of the App and its links as well as in arriving at conclusions. 

We have corrected the statement to read "LMIC users had the highest rate of in-app clicks for surgical peripheral nerve blocks" We agree that the specific subpopulations of providers will differ between LMIC and HIC, but it is not clear how to modify or adjust our presented findings on the basis of the size of the practitioner population and the distribution of the subtypes of providers (e.g. attendings, trainees, non-physicians). This difficulty led to our decision to provide unadjusted usage rates, which was the most straightforward way to interpret the incoming data. In addition, while we agree that educational tools should be tailored to individual needs, the design of these tools is beyond the scope of this article. However, this feature could be incorporated into future versions of this app. 

Reviewer 1, comment 4 :“Postoperative pain blocks were clicked at a higher percentage (2715/32.3%) in HIC compared to low (521/21.6%), lower middle (2090/21.4%), and 145 upper middle-income countries (2112/21.4%)”.

Maybe pertinent to consider the effect of confounding factors such as local policy relating to post operative pain relief that may have produced the above result.

The level of significance of the difference in App utilization between different income regions is not clearly explained.

Thank you for this comment. Given the limitations of this dataset (i.e., single app, Android platform), we did not perform any statistical analysis and present the data as purely descriptive. We agree with the reviewer that many factors may affect clinical practice at the institutional level and have added local policy as a confounding factor to the limitations of the study: Therefore, differences in participant income, lack of access to the internet or other unaccounted-for variables such as local policy or standards for analgesic regimens could skew the app user base and introduce confounding. 

Reviewer 2, comment 1: the data collection done can be interpreted as being one sided. The authors discuss how their collection of data may have answered their question, but they do fail to report on issues that may make the data collected less significant. Although they discuss the use of an app (that apparently just has links to youtube videos) they fail to simply comment and discuss the fact that it is an android app. One can assume that there is going to be a variance in the HIC to the LIC in terms of the use of android devices to non android devices. Does the app only work on android devices or can it be used on non-android devices as well? If that is the case, the introduction can lead to a little confusion. If it is that the app cannot, then the data might be slightly different if that unmentioned data is included. Meaning, those without androids how to learn and what are they searching for if looking up these videos on youtube directly? However that is not discussed and it is not entirely clear, in my eyes. In many HIC, like the U.S. and Eastern Europe, non-android devices may be utilized more frequently. Also, five years of data collection with today's technology may have data that may favor prior years to later years and vice-versa. 

Thank you for these comments. The app is solely offered for Android platforms, but Android applications are estimated to have 74% global market share penetration with higher use in LMICs. We now discuss the limitations to an Android only app. Please refer to our answer to Editors comment 3 for details.

Reviewer 2, comment 2: It is nice that the effect of COVID-19 is discussed, but is the data discovered simply because case loads were down or because people stopped looking or both or other? Would be interesting to comment about in the discussion. 

In a separate study we specifically sought to determine whether the anaesthesiology app data could serve as a qualitative proxy for global surgical case volumes and therefore had the ability to monitor the impact of the coronavirus disease 2019 pandemic. 

We added the following sentence to provide an explanation to the reviewers comment:

 In a separate study, App data provided a proxy for surgical case volumes, and could therefore be used as a real-time monitor of the impact of COVID-19 on surgical capacity [13]. 

11. Reviewer #3: well written ,insightful , the material is very relevant to current standards of practice in both academic and private group setting. I am recommending this material for publishing with no reservations

Thank you. 

Journal Requirements:

We ensured proper style and naming requirements.

2. We note that Figure 1 in your submission contain map images which may be copyrighted. All PLOS content is published under the Creative Commons Attribution License (CC BY 4.0), which means that the manuscript, images, and Supporting Information files will be freely available online, and any third party is permitted to access, download, copy, distribute, and use these materials in any way, even commercially, with proper attribution. For these reasons, we cannot publish previously copyrighted maps or satellite images created using proprietary data, such as Google software (Google Maps, Street View, and Earth). For more information, see our copyright guidelines: http://journals.plos.org/plosone/s/licenses-and-copyright.

This figure was created by one of the authors using the tmap package for R as described in Methods. 

We have updated the captions for Supporting Information at the end of the manuscript.

---

## [Decision Letter · Decision Letter 1]

24 Nov 2020

PONE-D-20-25898R1

Evaluating global interest in regional anesthesia educational resources based on income levels: an observational study of mobile health application data

PLOS ONE

Dear Dr. Moll,

Thank you for submitting your manuscript to PLOS ONE. After careful consideration, we feel that it has merit but does not fully meet PLOS ONE’s publication criteria as it currently stands. Therefore, we invite you to submit a revised version of the manuscript that addresses the points raised during the review process.

The authors have done a thoughtful job revising the manuscript.  It now informs the readers of the strengths and limitations to this study.  However, the reviewer makes an important point regarding the title.  Please change to reflect the international yet limited nature of this survey.

We look forward to receiving your revised manuscript.

Kind regards,

Mercedes Susan Mandell, MD PhD

Academic Editor

PLOS ONE

Reviewers' comments:

Reviewer's Responses to Questions

**Comments to the Author**

1. If the authors have adequately addressed your comments raised in a previous round of review and you feel that this manuscript is now acceptable for publication, you may indicate that here to bypass the “Comments to the Author” section, enter your conflict of interest statement in the “Confidential to Editor” section, and submit your "Accept" recommendation.

Reviewer #1: (No Response)

2. Is the manuscript technically sound, and do the data support the conclusions?

Reviewer #1: Partly

3. Has the statistical analysis been performed appropriately and rigorously? 

Reviewer #1: I Don't Know

4. Have the authors made all data underlying the findings in their manuscript fully available?

Reviewer #1: No

5. Is the manuscript presented in an intelligible fashion and written in standard English?

Reviewer #1: Yes

6. Review Comments to the Author

Reviewer #1: 1. The 'Changed' title still suggests that this was an evaluation of GLOBAL trends though the content does not support that assertion. It is essentially data collected form LMICs and HICs unless it is proven that the sample is representative of global usage patterns!

2. It is unclear how it could be asserted that "LMIC users has the highest rate of in-qpp clicks for surgical peripheral blocks' when it is unclear how the rate is accurately calculated.

3. The effect of confounding factors on the choice of blocks has been addressed.

7. PLOS authors have the option to publish the peer review history of their article (what does this mean?). If published, this will include your full peer review and any attached files.

Reviewer #1: **Yes: **M.B.Gunetilleke

---

## [Author Response · Author response to Decision Letter 1]

28 Nov 2020

Dear Professor Mandell, 

Thank you for the opportunity to further revise and clarify our manuscript. As requested, we are providing answers to the reviewers comments and suggestions below. 

1. The 'Changed' title still suggests that this was an evaluation of GLOBAL trends though the content does not support that assertion. It is essentially data collected form LMICs and HICs unless it is proven that the sample is representative of global usage patterns!

 We changed the title to reflect the reviewers suggestion to “Differential interest in regional anesthesia educational resources based on country income level: an observational study of mobile health application data”. 

2. It is unclear how it could be asserted that "LMIC users has the highest rate of in-app clicks for surgical peripheral blocks' when it is unclear how the rate is accurately calculated.

 The original Reviewer indicated:

Reviewer 1, comment 3: “LMIC users had the highest in-app clicks for surgical peripheral nerve blocks” Since the results may be used for designing educational tools based on the unique requirements of each group, it might be useful to consider size of practitioner population in each of these categories (LMIC, HIC) and give appropriate weightage when considering the utilization of the App and its links as well as in arriving at conclusions.

 In our original response to the reviewer, we wrote:

We have corrected the statement to read "LMIC users had the highest rate of in-app clicks for surgical peripheral nerve blocks" We agree that the specific subpopulations of providers will differ between LMIC and HIC, but it is not clear how to modify or adjust our presented findings on the basis of the size of the practitioner population and the distribution of the subtypes of providers (e.g. attendings, trainees, non-physicians). This difficulty led to our decision to provide unadjusted usage rates, which was the most straightforward way to interpret the incoming data. In addition, while we agree that educational tools should be tailored to individual needs, the design of these tools is beyond the scope of this article. However, this feature could be incorporated into future versions of this app.

Concerns with the accuracy of the rate of in-app click calculation were not elucidated in the initial review, and it is difficult to respond to the reviewer concern here without a specific reason for the reviewer concern now. The rate of in-app clicks was, as indicated, an unadjusted in-app click rate (numerator: raw subtype click; denominator: raw overall click). However, out of an abundance of caution in clarity of meaning, we have revised the sentence (second sentence of Discussion) to more closely mirror the one following it:

“With in-app clicks as the proxy measure, LMIC users demonstrated greater interest in surgical peripheral nerve blocks (ankle block, axillary, infraclavicular, interscalene, supraclavicular blocks) as compared to users from HIC.”

3. The effect of confounding factors on the choice of blocks has been addressed.

 Great. Thank you.

Thank you. 

Respectfully,

Vanessa Moll, MD, PhD, FCCM, FASA

---

## [Editor Report · Decision Letter 2]

1 Dec 2020

PONE-D-20-25898R2

Differential interest in regional anesthesia educational resources based on country income level: an observational study of mobile health application data

PLOS ONE

Dear Dr. Moll,

Thank you for submitting your manuscript to PLOS ONE. After careful consideration, we feel that it has merit but does not fully meet PLOS ONE’s publication criteria as it currently stands. Therefore, we invite you to submit a revised version of the manuscript that addresses the points raised during the review process.

The authors have modified the title.  Can they please consider a title that is more focused.  The current suggestion could be improved.  Would they consider something such as "The use of mobile anesthesia applications for clinical practice in countries with a lower gross national product".

We look forward to receiving your revised manuscript.

Kind regards,

Mercedes Susan Mandell, MD PhD

Academic Editor

PLOS ONE

---

## [Author Response · Author response to Decision Letter 2]

3 Dec 2020

Dear Professor Mandell,

thank you for the opportunity to fine tune the title. We have now changed the title to “Regional anesthesia educational material utilization varies by World Bank income category: a mobile health application data study.” The inclusion of the type of data analyzed is keeping with current reporting guidelines. The title is now 17 words long (previously 20). 

We think that this title is accurately reflecting the study and hope that the reviewers will find the changes satisfactory.

Respectfully,

Vanessa Moll, MD, PhD, FCCM, FASA

---

## [Editor Report · Decision Letter 3]

18 Dec 2020

Regional anesthesia educational material utilization varies by World Bank income category: a mobile health application data study

PONE-D-20-25898R3

Dear Dr. Dr. Moll,

We’re pleased to inform you that your manuscript has been judged scientifically suitable for publication and will be formally accepted for publication once it meets all outstanding technical requirements.

Kind regards,

Mercedes Susan Mandell, MD PhD

Section Editor

PLOS ONE
---

## [Editor Report · Acceptance letter]

21 Jan 2021

PONE-D-20-25898R3 

Regional anesthesia educational material utilization varies by World Bank income category: a mobile health application data study 

Dear Dr. Moll:

I'm pleased to inform you that your manuscript has been deemed suitable for publication in PLOS ONE. Congratulations! Your manuscript is now with our production department. 

Kind regards, 

on behalf of

Dr. Mercedes Susan Mandell 

Section Editor

PLOS ONE